# Used Nasogastric Feeding Tubes from Neonates Contain Infant-Specific Bacterial Profiles

**DOI:** 10.3390/microorganisms11061365

**Published:** 2023-05-23

**Authors:** Sandra Meinich Juhl, Karen Angeliki Krogfelt, Witold Kot, Dennis Sandris Nielsen, Lukasz Krych

**Affiliations:** 1Department of Neonatology, Rigshospitalet–Copenhagen University Hospital, 2100 Copenhagen, Denmark; 2Department of Virus and Microbiological Special Diagnostics, Statens Serum Institut, 2300 Copenhagen, Denmark; 3Department of Science and Environment, Roskilde University, 4000 Roskilde, Denmark; 4Department of Plant and Environmental Sciences, University of Copenhagen, 1871 Frederiksberg C, Denmark; 5Department of Food Science, Faculty of Science, University of Copenhagen, Rolighedsvej 26, 1958 Frederiksberg C, Denmark; dn@food.ku.dk (D.S.N.); krych@food.ku.dk (L.K.)

**Keywords:** infant, neonatal intensive care unit, infection risk, feeding tube, contamination, colonization

## Abstract

Nasogastric feeding tubes (NG-tubes) from neonates contain potentially pathogenic bacteria. Using culture-based techniques, we have previously determined that the usage duration of NG-tubes did not impact the colonization of the nasogastric tubes. In the present study, we performed 16S rRNA gene amplicon sequencing to evaluate the microbial profile of 94 used nasogastric tubes collected from a single neonatal intensive care unit. Using culture-based whole genome sequencing, we as-sessed whether the same strain persisted in NG-tubes collected from the same neonate across different time-points. We found that the most commonly occurring Gram-negative bacteria were *Enterobacteriaceae*, *Klebsiella* and *Serratia*, while the most common Gram-positive bacteria were staphylococci and streptococci. The microbiota of the NG-feeding tube was overall infant-specific, rather than dependent on the duration of use. Furthermore, we determined that reoccurring species from the individual infant represented the same strain and that several strains were common for more than one infant. Our findings indicate that bacterial profiles found in NG-tubes of neonates are host-specific, not dependent on the duration of use and strongly influenced by the environment.

## 1. Introduction

At birth, the newborn infant is immediately confronted with immense microbial pressure from the outside world, leading to colonization of the mucosal membranes and the skin. When an infant is admitted to a neonatal intensive care unit (NICU), their natural colonization pattern is compromised compared to newborns placed in the arms of the mother [1]. The main factors influencing the colonization pattern include the use of antibiotics, less skin-to-skin contact, contact with different caregivers and the surfaces of the NICU, along with the caregiving equipment. This may ultimately lead to dysbiosis [2]. In very vulnerable preterm infants, dysbiosis has been shown to increase the risk of developing necrotizing enterocolitis (NEC) [3,4,5].

Nasogastric tubes (NG-tubes) used to ensure nutritional intake in sick neonates and preterm infants are indwelling foreign bodies, and biofilm formation occurs within 24 h of placement [6]. The NG-tubes are exposed to the microbiota of the nose, throat and gut of the infant during their placement. Aspiration of the gut content into the NG-tube is performed immediately after placement to ensure its correct positioning. This may introduce intestinal microbiota into the NG-tube. Potential pathogens, such as *Enterococcus faecalis*, *Enterobacter cloacae*, *Serratia marcescens* and *Klebsiella pneumoniae*, have all previously been found in NG-tubes from neonates [7,8]. The NG-tubes may at any point be exposed to other bacteria when handled by caregivers and when the external part of the NG-tube is lying in the nest next to the infant between feeds. Inspired by a quality-improvement study suggesting that weekly renewal of NG-tubes might have been a contributing factor to an observed decrease in NEC incidence [9], we examined used NG-tubes collected from infants admitted in our NICU [10]. We found a high incidence of contamination of feeds given through the NG-tubes, but no correlation between the duration the NG-tube had been in use and the quantity or the type of bacteria that were inadvertently given to the infant when fed through the NG-tube. Thus, the biofilm in the NG-tube seemed to be formed immediately after the first feedings. However, we also found that 89% of the NG-tubes contained bacteria, and half of them were potential pathogens. This could occur within less than one day of use [10].

Hence, we wondered if the biofilm consisted mainly of bacteria from the stomach (which were pulled into the feeding tube when gastric content was aspirated to check for residuals), from the environment or from the milk fed via the NG-tube.

In the present study, we performed a detailed molecular analysis of the microbiota collected from NG-tube flushes from our previous work [10] to compare the patterns of bacteria composition in the NG-tubes and correlate them to the specific infants. Furthermore, bacterial isolates from infants with four or more collected NG-tubes were whole genome sequenced for strain level discrimination. To our knowledge, no other studies have characterized the bacteria found in consecutive NG-tubes from the same neonate to the strain level.

## 2. Materials and Methods

### 2.1. Study Population

This prospective and observational study was conducted at the neonatal intensive care unit, Rigshospitalet, Copenhagen, Denmark. The inclusion criteria of enrolled infants were (1) admission to the neonatal department and (2) a resident NG-tube at the time of inclusion. The methods related to feeding tube collection and culturing were previously described [10]. In short, the trial lasted two months, between April and June 2014. In total, 94 used Kangaroo™ polyurethane feeding tubes (Covidien, Dublin, Ireland, NG-tubes) were collected for analysis from 34 infants admitted to the department. Between one and nine NG-tubes per infant were collected (median = two). Baseline data regarding infants and NG-tubes presented in the previous study [10] are summarized in Appendix A. The NG-tubes were exchanged weekly or more often whenever necessary (if clotted or misplaced). The infants were given raw mother’s milk as the first choice, and if insufficient in amount, it was combined with pasteurized donor milk or preterm formula. Nurses, as well as parents, were involved in NG-tube feeding. Nurses wore gloves, and parents were instructed in careful hand hygiene. Infants with gestational ages (GA) < 30 weeks were given probiotics daily (2 capsules of 10^9^ CFU *Lactobacillus rhamnosus* GG (LGG) + 10^8^ CFU *Bifidobacterium animalis* ssp. *lactis* (BB12)) from the third day of life.

### 2.2. Ethics

The study was approved by the Danish Ethical Committee (Protocol number: H-1-2014-009), and written informed consent was sought from one of the parents according to Danish law.

### 2.3. Bacteriological Analyses

The used NG-tubes were flushed with 1 mL NaCl to mimic a 1 mL meal given through the NG-tube. The flush was analyzed with both molecular and culture-based methods. Bacterial isolates were cultured and identified as previously described [10]. Unique colonies were counted, re-streaked until purity, identified using Maldi-Tof and stored at −80 °C until further use.

### 2.4. Bacterial Profiling with High-Throughput Sequencing

The bacterial community of 94 NG-tubes was determined using tag-encoded 16S rRNA gene amplicon (V3 region), NextSeq-based (Illumina, San Diego, CA, USA), high-throughput sequencing. Cellular DNA was extracted from 200 μL of the flush from the feeding tube using a PowerSoil DNA Isolation Kit (Mo Bio Laboratories, Carlsbad, CA, USA) according to the manufacturer’s instructions, including an initial bead-beating step using FastPrep-24™ 5G (MP Biomedicals, Santa Ana, CA, USA). The DNA concentration was evaluated using a Qubit^®^ dsDNA HS Assay Kit (Life Technologies, Carlsbad, CA, USA). The measurement was performed using a Varioskan Flash Multimode Reader (Thermo Fischer Scientific, Waltham, MA, USA). Fluorescence was measured at 485/530 nm. The DNA concentration was normalized to 10 ng/μL and subjected to library preparation using two-step PCR.

The V3 region (~190 bp) of the 16S rRNA gene was amplified using the primers nxt388_F: (5′-TCGTCGGCAG CGTCAGATGT GTATAAGAGA CAGACWCCTA CGGGWGGCAGCAG-3′) and nxt518_R: (5′-GTCTCGTGGG CTCGGAGATG TGTATAAGAG ACAGATTACC GCGGCTGCTGG-3′), compatible with the Nextera Index Kit (Illumina, San Diego, CA, USA). The first PCR round was performed in 25 μL of final reaction volume, composed of 12 μL of AccuPrimeTM SuperMix II (Life Technologies, CA, USA), 0.5 μL of each primer (10 μM, nxt388_F and nxt518_R), 5 μL of genomic DNA (~10 ng/ul) and nuclease-free water up to 25 μL. The PCR1 temperature profile was as follows: 95 °C for 2 min, followed by 35 cycles of 95 °C for 15 s, 55 °C for 15 s and 68 °C for 40 s. Lastly, a final extension step was held in 68 °C for 5 min (Agilent SureCycler 8800 termocycler, Santa Clara, CA, USA).

In order to incorporate sample-specific barcodes, a second PCR round was performed containing 12 μL of Phusion High-Fidelity PCR Master Mix (Thermo Fisher Scientific, Waltham, MA, USA), 2 μL of the corresponding P5 and P7 primers (Nextera Index Kit, Illumina, San Diego, CA, USA), 2 μL of the PCR1 product and nuclease-free water to a final volume of 25 μL. The PCR2 temperature profile was as follows: 98 °C for 1 min, followed by 12 cycles of 98 °C for 10 s, 55 °C for 20 s and 72 °C for 20 s. A final extension step was held in 72 °C for 5 min (Agilent SureCycler 8800 termocycler, Santa Clara, CA, USA). The amplified fragments with adapters and tags were purified using AMPure XP beads (Beckman Coulter Genomic, Brea, CA, USA), according to the manufacturer’s manual. Each PCR round was conducted in the presence of positive (*E. coli* genomic DNA) and negative controls (a no template control (NTC) and a negative extraction control). Prior to libraries pooling, clean constructs were quantified using a Qubit fluorometer (Invitrogen, Carlsbad, CA, USA), and finally, PCR2 products were mixed equimolarly. A single flow cell of a 150 bp pair-ended NextSeq (Illumina, San Diego, CA, USA) sequencing run was performed according to manufacturer’s instructions.

### 2.5. Whole Genome Sequencing (WGS) of Bacterial Isolates

To determine whether a specific strain of bacteria occurred repeatedly in each infant’s NG-tube, we investigated the NG-tubes of nine extremely preterm infants, from whom four or more NG-tubes were collected. In 8 of the 9 infants who fulfilled this criterion, the same bacterial species reoccurred in 50% or more of the NG-tubes (Table 1). The reoccurring bacterial colonies (total of 47) were subjected to WGS. The data from two isolates were rejected due to a low sequencing yield. See Appendix A for details on colonies’ identification. Genomic DNA from bacterial colonies was extracted using the DNeasy Blood and Tissue kit, as described by the manufacturer (Qiagen, Valencia, CA, USA). The DNA concentration was evaluated using the Qubit^®^ dsDNA HS Assay Kit (Life Technologies, CA, USA), as described above. The DNA concentration was normalized to 1 ng/μL and subjected to library preparation using the Nextera XT DNA Library Preparation Kit (Illumina, San Diego, CA, USA), according to the manufacturer’s instruction, and sequenced with the NextSeq500 using Mid Output Kit v2.5 (300 cycles, #20024905, Illumina, San Diego, CA, USA). The 16S rRNA gene amplicon library and WGS library were sequenced simultaneously on the NextSeq500 platform in 1:1 proportion.

### 2.6. Data Analysis

#### 2.6.1. 16S rRNA Gene (V3 Region) Amplicon Sequencing

The raw dataset containing pair-ended reads with corresponding quality scores were merged and trimmed using fastq_mergepairs and fastq_filter scripts implemented in the VSEARCH pipeline [11], using the following settings: -fastq_minovlen 100, -fastq_maxee 2.0, -fastq_truncqual 4, -fastq_minlen 150. Purging the dataset from chimeric reads and constructing Amplicon Sequence Variants (ASV) was conducted using UNOISE3 within the VSEARCH pipeline [11]. The Greengenes (13.8) 16S rRNA gene collection was used as a reference database [12]. The Quantitative Insight Into Microbial Ecology (QIIME) open-source software package (1.9.0) was used for subsequent analysis steps [13]. Exactly 1 sample was removed due to a low read number (bellow 95,000 reads/sample, n = 79,096). UniFrac distance matrices were generated with the Jackknifed Beta Diversity workflow, based on 10 distance metrics calculated using 10 subsampled ASV-tables. The number of sequences taken for each jackknifed subset was set to 85% of the sequence number within the most indigent sample (95,000 reads/samples). Permutational Multivariate Analysis of Variance (PERMANOVA, compare_categories.py, QIIME 1.9.0) was used to evaluate group differences based on weighted and unweighted UniFrac distance matrices. Alpha diversity expressed with an observed species index (lowest possible level of ASV classification) was computed for rarefied ASV-tables (95,000 reads/sample) using the alpha rarefaction workflow (alpha_diversity.py, QIIME 1.9.0). Differences in alpha diversity were determined using a *t*-test-based approach employing the non-parametric (Monte Carlo) method (999 permutations) implemented in the compare alpha diversity workflow (compare_alpha_diversity.py, QIIME 1.9.0).

#### 2.6.2. WGS of Bacterial Isolates

The raw reads were trimmed from adaptors, barcodes and low-quality fragments using Trimmomatic v0.35 [14]. The high-quality reads (Q score 15, allowing for no more than 3% chance for wrong basecalling) with a minimum size of 50 bp were retained for further analysis. For every sample, assembly of the raw read was carried out using SPAdes v3.5.0 [15], and contigs with a minimum length of 10,000 bp were subjected to classification with kraken2 [16] against the NCBI database [17] and Average Nucleotide Identity using orthologous fragment pairs (OrthoANI) [18]. The taxonomic classification with kraken2 was selected based on the highest confidence scores, taxonomic hierarchy and biological context of samples.

#### 2.6.3. Data Availability

Raw sequencing data (fastq) for the 16S rRNA gene amplicon sequencing and WGS data of 45 isolates were deposited in the NCBI SRA database (https://www.ncbi.nlm.nih.gov/sra, accessed on 6 April 2023) under the Bioproject: PRJNA951070.

## 3. Results

Flushes from 94 nasogastric feeding tubes from 34 infants over the period from April to June 2014 were microbiologically characterized. The infants had a median GA of 30.1 weeks and a median birth weight of 1083 g. The NG-tubes had been in place for a median of 3.25 d (range 8 h–14.2 d), and the postnatal median age at the time of collection was 37 d (range = 1–119 d). The details regarding culture-specific findings have been previously described (See Table 2 of the original paper) [10].

### 3.1. Bacterial Profiling with High-Throughput Sequencing

The bacterial community determined with tag-encoded 16S rRNA gene (V3 region) high-throughput sequencing indicated high level of similarity between the microbial community of NG-tubes that originated from the same infant (Figure 1). The NG-tubes of 9 neonates were dominated by Gram-negative (GN) bacteria, whereas the remaining 23 tubes had a higher prevalence of Gram-positive (GP) bacteria (Figure 1A,B). The most commonly occurring GNs were identified as belonging to *Enterobacteriaceae*, *Klebsiella* and *Serratia* spp., and the most common GP bacteria were staphylococci and streptococci. We also observed the probiotic bacteria *Bifidobacterium animalis* and different types of lactobacilli in several samples (Figure 1). The PERMANOVA test performed on data from infants from whom four tubes or more were obtained showed significant clustering according to the infant, based on both unweighted and weighted UniFrac distance matrices (Figure 2A,B). The PERMANOVA analysis based on weighted or unweighted UniFrac distance metrices showed no significant clustering due to the NG-tubes’ usage duration (Figure 2C,D) or feed volumes nor antibiotics usage. Finally, no significant differences were reported in alpha diversity (observed species index) between the tested categories reflecting the time ranges of NG-tubes usage duration (Figure 2E).

### 3.2. Whole Genome Sequencing

WGS showed that isolates classified as *Klebsiella oxytoca*, *Enterococcus faecium* and *Staphylococcus epidermidis* showed a high level of OrthoANI similarity, ranging between 99.7% and 100%, characteristic of being similar at a strain level (Figure 3). Among isolates classified as *Enterobacter cloacae*, 2 clusters of isolates with a within-cluster similarity above 99.9% were identified. The average difference in OrthoANI indices between these 2 clusters reached 9%. This could indicate the presence of two strains within this category. One of the *E. cloacae* strains was isolated from NG-tubes belonging to a single infant, whereas the other strain was present in tubes from two other infants, who were siblings and, hence, fed with the same maternal milk.

## 4. Discussion

In this study of bacteria from used NG-tubes of neonates admitted to a neonatal intensive care unit, we found that the bacterial profile of each NG-tube was infant-specific rather than dependent on the duration of use. Furthermore, we determined with WGS that reoccurring species from the individual infant represented the same strain. Interestingly, the same bacterial strains occurred in more than one infant.

Strains of the *Staphylococcus epidermidis*, *Enterococcus faecium*, *Klebsiella oxytoca* and *Enterobacter cloacae* species were identified from NG-tubes from one neonate over time, but also from other neonates in the same period. All three bacterial species have previously been reported as common microbial members residing on human skin and in hospital environments [19]. *Klebsiella oxytoca* and *Enterococcus* are often resistant to antibiotics and, therefore, thrive in hospitals [20], whereas *Staphylococci* and *Enterococci* are often human contaminants from the skin, but all can lead to sepsis in preterm infants [21].

Our data suggest that bacteria found in the NG-tubes of neonates may derive from the stomach of the infant when residuals are drawn into the NG-tube before feeding, but they also indicate some bacteria are derived from a common source by sharing the environment. The NICU environment and the gut of admitted infants have previously been shown to contain the same bacterial strains [1], and there may even be a specific room microbiota, by which the infant may be colonized when admitted to the NICU [22]. The findings of this study are consistent with our previous results, in which by using culture-dependent methods, we confirmed that the NG-tubes usage duration did not affect the abundance or identity of bacteria found across tested samples. Our group recently published a small study of 22 preterm infants, in which we investigated if changing NG-tubes daily (in the first week of life), compared to weekly exchange, affected the microbiota of the gastric aspirate on day 7 [23]. Surprisingly, in both investigated groups, low bacterial counts were reported, which suggested that frequent exchange of NG-tubes is not necessary to prevent the incidence of contamination. Since gastric aspirates collected immediately after birth contain nearly no bacteria [24], it was concluded that NG-tube colonization mainly derives from the microbiota of the infant and the environment. Finding the same bacterial strains in different infants strongly suggests that the gastrointestinal tract of those infants are colonized with a common source in the NICU microbiota, and afterwards, the NG-tubes are colonized with the bacteria from the gastrointestinal content of the infant.

When using culture-dependent methods, only limited proportions of probiotic bacteria were found (*B. animalis* was detected in 1/94 samples, whereas *L. rhamnosus* was detected in 14/94 samples). Although 16S rRNA gene amplicon sequencing has disclosed probiotics in more samples than the culture-dependent methods, it is unknown whether these were viable bacteria.

Furthermore, it is important to stress that since the WGS data rely on culture-dependent methods, our conclusions on bacterial taxonomy, presence/absence and relative abundance are limited to bacteria that favored the specific growing conditions.

The ultimate clinical question is whether bacteria of NG-tubes are potential pathogens for the infant. For immune-competent individuals with a low pH in the stomach, they may not be a big threat, but this may be different for the preterm population. Previous studies showed that once a pathogen occurs in a blood stream infection of one infant, it is likely to occur in another infant in the same unit [25]. NG-tubes may serve as a reservoir for potential pathogens that can spread to other infants in the department. However, as they are necessary for supplying the infant with nutrition, it may be difficult to change practice. If the bacteria of the NG-tubes are mainly mirroring the microbiota of the gastric content, there may not be a problem at all. This would suggest that changing the frequency of exchanging the NG-tubes is not necessary to prevent contamination with pathogens.

This was a follow-up on a previous study, where culture-based methods were used to determine the bacterial load from NG-tubes. The present study was the first study using molecular methods to determine the microbiota of used NG-tubes and relate it to the duration of use. This was a convenience study, as we collected all used NG-tubes in our department for a limited period, thus giving us a broad study group, both term and preterm. Infants born <30 weeks GA received probiotics, which may have affected the colonization of the feeding tube in these infants. In the future, it may be interesting to investigate which factors are most likely to determine which infants end up with the same bacteria in their NG-tubes, e.g., sharing the room, sharing caregivers and so forth. For that, further research is required to test other potential sources of bacterial transmission, such as, for example, the diaper zone, skin or breast milk. By creating more knowledge in this field, we will be able to understand how microbiota is shared between hospitalized infants and prevent transmission of pathogenic bacteria.

## 5. Conclusions

The microbiota present in the NG-tubes of infants in a neonatal department seemed to be infant-specific, rather than dependent on the duration of use. The bacteria may originate from the microbiota of the stomach of the infant itself when gastric aspirates are pulled into the tube before a meal is given. However, the same bacterium was present in the NG-tubes of different infants, indicating that there may be a common source for the colonization, which could be the caretakers or the NICU environment.

## Figures and Tables

**Figure 1 microorganisms-11-01365-f001:**
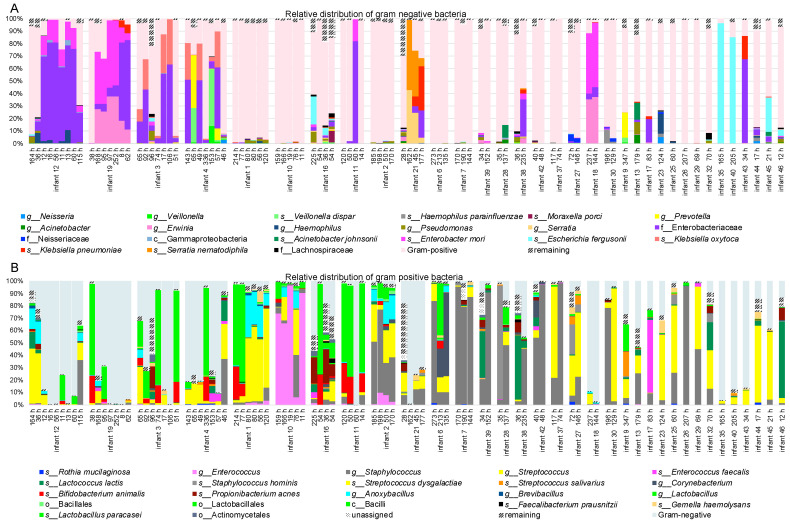
Relative abundance of bacterial community inhabiting feeding tubes. Bar-chart presenting bacterial relative distribution of most abundant taxa determined with 16S rRNA gene amplicon sequencing (NextSeq, Illumina) with focus on Gram-negative (**A**) and Gram-positive bacteria (**B**). Ten infants had their tube replaced four times or more. Remaining infants had between one and three tubes. The time at which given tube was exchanged is listed in the brackets (h). The analysis indicates that presence of pathogenic and potential pathogenic bacteria is rather infant-specific feature rather than being conditioned by time that given feeding tube is used. Category “unassigned” represents reads that found no hits in the Greengens database. Category “reaming” represents summarized low-abundant taxa (bellow 0.5% abundance across all samples).

**Figure 2 microorganisms-11-01365-f002:**
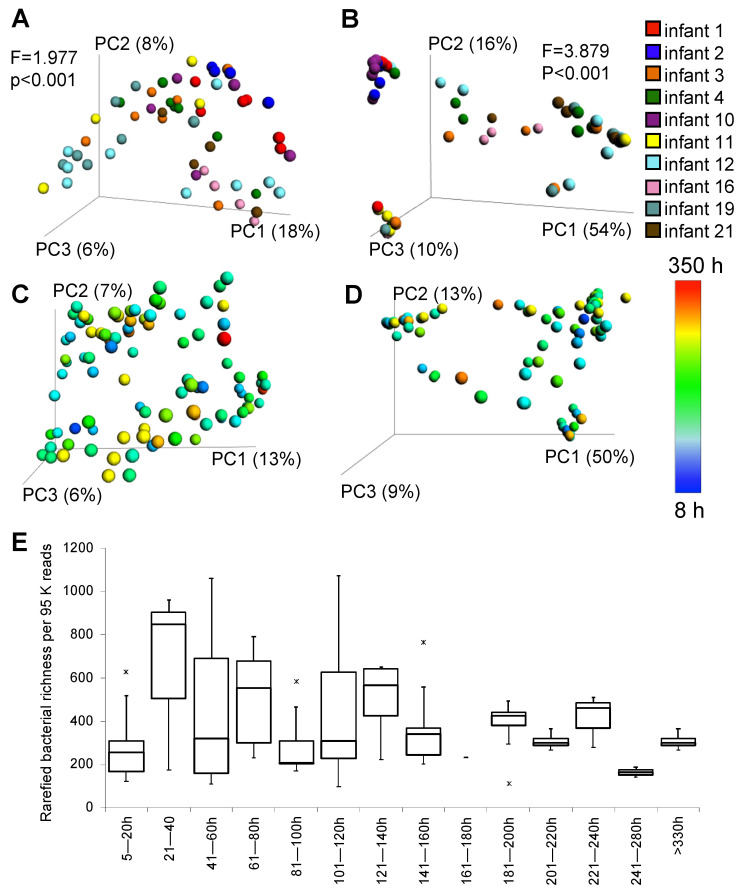
Alpha and beta diversity analysis of the microbial composition. Principal Coordinates Analysis (PCoA) plots based on unweighted (**A**) and weighted (**B**) UniFrac distance matrices depicting NG-tubes-associated microbial composition collected from infants with multiple replacements of NG-tubes (four or more). PERMANOVA results given in the plot area indicate that the microbial composition is infant-specific. PCoA plots based on unweighted (**C**) and weighted (**D**) UniFrac distance matrices demonstrating that the duration of feeding tube application did not influence qualitative nor quantitative differences in microbial composition. Similarly, no clear tendency in variation of species reaches index could be observed depending on the duration of feeding tube application (**E**) “x” symbol indicates the Min and Max outliers.

**Figure 3 microorganisms-11-01365-f003:**
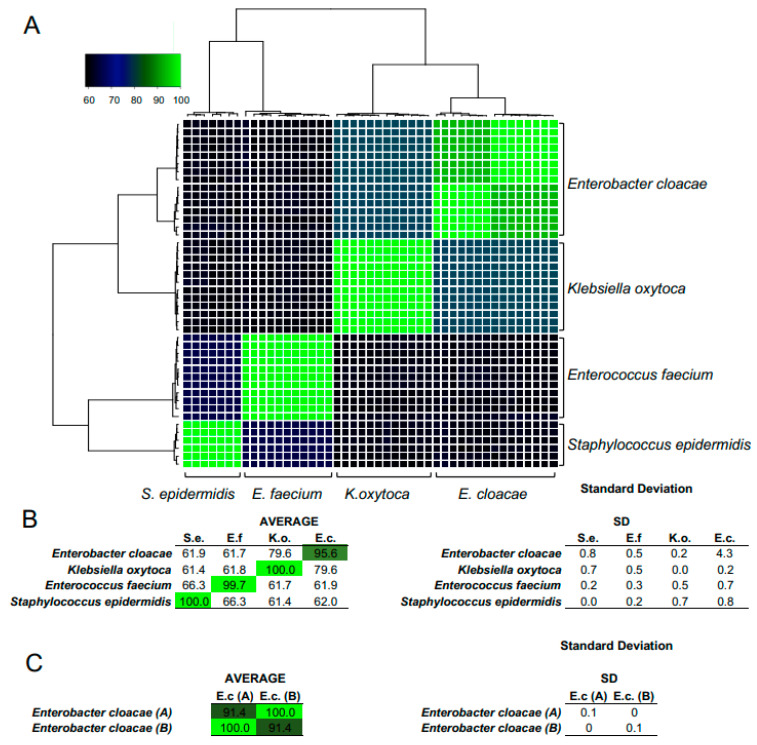
Whole genome sequencing comparison of bacterial isolates. Heatmap presenting comparison of average nucleotide indices using orthologous fragment pairs (OrthoANI) (**A**). The total of 47 isolates were subjected to whole genome sequencing (NextSeq, Illumina). SPAdes assembled contiguously were used for classification with kraken2 and for OrthoANI indices comparison. Isolates classified to *Kliebsella oxytoca, Enterococcus faecium* and *Staphylococcus epidermidis* showed high-level similarity ranging between 99.7% and 100% (**B**). Among isolates classified as *Enterobacter cloacae*, 2 clusters of similarity above 99.9% were identified. The average difference in OrthoANI indices between these 2 clusters reached 9% (**C**). This could indicate the presence of two *Enterobacter cloacae* strains within this category.

**Table 1 microorganisms-11-01365-t001:** Nine infants had four or more nasogastric feeding tubes collected. Bacteria cultured in > 50% of the collected nasogastric tubes are shown here. ^a^ Infants 5–7 were triplets.

Patient ID	Birth Weight (g)	Gestational Age (Week)	Species Cultured in >/= 50% of NG-Tubes	Found in x of y NG-Tubes (x/y)
1	580	24 + 5	*Staphylococcus epidermidis*	4/6
2	570	26 + 4	*Staphylococcus epidermidis* *Enterococcus faecium*	4/4 3/4
3	815	25 + 3	*Klebsiella oxytoca*	6/7
4	910	26 + 6	*Klebsiella oxytoca*	6/7
5 ^a^	790	25 + 1	*Enterococcus faecium*	5/5
6 ^a^	565	25 + 1	*Enterococcus faecium* *Enterobacter cloacae*	3/4 2/4
7 ^a^	800	25 + 1	*Enterobacter cloacae*	7/9
8	810	26 + 2	None	
9	650	24 + 4	*Enterobacter cloacae*	7/7

## Data Availability

Publicly available datasets were analyzed in this study. This data can be found here: https://www.ncbi.nlm.nih.gov/sra under the Bioproject: PRJNA951070 accessed on 6 April 20223.

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
