# Peer review of "Used Nasogastric Feeding Tubes from Neonates Contain Infant-Specific Bacterial Profiles"

_microorganisms, 2023, doi:10.3390/microorganisms11061365_

Round 1

Reviewer 1 Report

The authors have made a commendable effort to address an important question in preterm infant microbiome acquisition - "How do bacteria within indwelling NG-tubes influence infant microbiomes".

The use of WGS enables classification of bacteria with strain-level resolution which is greater than many previous studies have achieved. 

The authors note that this study has been performed off the back of previous studies utilising these samples. This is not an issue in principle as the inclusion of strain level analysis and identification of persistent strains within single infants despite NG-tube replacement is a novel finding. However, it is this reviewer's belief that the analysis and presentation of data in the current manuscript is not sufficient for publication. The methods are not written to be reproducible. Very little comparison to or contextualisation  against previous research is performed and figures do not appear publication quality.

Please address comments included below: 

General comments:

Please check citations. Only 18 seems very low for manuscripts conducting so many experiments. Similarly, it appears some citations are irrelevant to the statements made (e.g. reference 3 in introduction, reference 7 in methods.) Others have already explored the bacteria colonising NG tubes. Could the authors cite these papers too?

The paper could definitely benefit from a re-write. Sentences frequently make no sense. 

Introduction:

Second paragraph in the introduction feels far more like methods for previous studies than an introduction. Could this be concentrated down to "Previous research from our group has suggested X". This would enable the authors to provide more in-depth background to the importance of NG tube colonisation and wider preterm infant microbiome acquisition and development.

Methods:

Please outline number of infants enrolled and number of NG tubes samples per infant.

Molecular methods used to explore NG tubes are not explained in citation 6. Please provide further information in text (e.g.  primers or library prep kit used).

How many of 94 NG-tubes were from the same infant? How many infants provided 4 or more NG tubes?

"Bacterial species occurring in more than 50 % of the 98 NG-tubes from the same individual were subjected to WGS" -  Use of culturomics to sequence bacterial species isolated limits the strains identified to those able to grow on utilised media. This could be referenced in the discussion as a limitation of the study.

Methods are not reproducible from the explanation provided in the data analysis section. Could the authors provide more information on tools used and settings for sequence processing (e.g. specific processes performed in QIIME). While the authors cite their own previous studies these are not sufficient to enable reproduction.  

"using the UNOISE " - Seems a word is missing here?

Please explain threshold for low read sample exclusion. How many reads were required for samples to be included in the study?

Were read counts per samples normalised prior to calculating beta diversity metrics?

Were any kit or sequencing negative controls included in 16S rRNA experiment? Please describe utilisation of controls and how you validated legitimacy of bacterial taxa identified in samples.

Are sequencing data publicly available in any sequence repository? This would vastly increase confidence in the data presented and improve reproducibility.

Were all assembled contigs classified with Kraken2? Was any clustering or deduplication performed to identify representative sequences? Assuming several different classifications were assigned by Kraken2 how did the authors decide which was representative of the original bacterial isolate? 

Results:

Please provide a table or figure to show the spread of patient associated metadata (GA, BW, DOL, NG-days etc.).

Could the authors perform a PERMANOVA to illustrate the high level of similarity between microbial communities of NG-tubes from the same infant? How does this compare to the effect of infant age at sampling, feed volumes, exposure to antibiotics etc?

Figure 1 - axis labels are hard to read - could these be adjusted to make clearer. Assuming bars are grouped by patient? This is not explained in legend. Why are gram+ and gram- reported seperately? It would be useful to see how proportions of Staph, Strep, Enterococcus etc. compare to Enterobacteria. What does "remaining" mean in the legend?

"We observed no clear effect of the duration the NG-tubes had been used with respect to relative abundance or species diversity" - Sentence doesn't make sense. Please revise.

Please list results of significance testing in text. Why is ANOSIM performed to quantify the similarity between samples from a single infant but not NG-tube duration.

Figure 2 - Y Axis title should be "rarefied bacterial richness". Is this performed at the species level? Most bacterial annotations from Figure 1 are not at species level.

"The reoccurring bacterial colonies (total of 47)" - Please provide details of all bacterial colonies identified as supplementary material. This will be very interesting for others working in this field. E.g. Did no probiotic strains colonise the NG-tubes? Could failure to ID these be due to the culturomics approaches used?

What were the assembled contig lengths of isolates interrogated with OrhoANI? Could high similarity be due to low contig length spanning highly conserved regions of genomes?

Figure 3: - "SPAdes assembled contiguous" should be "SPAdes assembled contigs" or "SPAdes assembled contiguous reads"

- No description of table headers in figure. Should these tables be included in the figure? Should these not be supplementary tables?

Discussion:

The authors could provide much greater contextualisation in respect to previous studies and how these results may influence NG-tube use in clinical practice. What is the importance of finding these microbes in NG-tubes? Were they also identified as sepsis causing organisms at similar times as samples were taken, in the same NICU? 

Reviewer 2 Report

I value the paper, and the novelty of the approach taken. As a general reflection, it reads still somewhat that tube colonization is an unique event, while other sites have been described. In my opinion, there would be value to somewhat elaborate more on NICU colonization patterns to put the current findings into perspective.

In the abstract, the authors suggest a common source in the environment. As it is not clear if sampling has been done in one specific NICU, I would suggest to add this to the abstract (as the environment is likely unit, or region dependent).

Is it fair to state that the NG colonization is in essence not different for other sites to swap in neonates (diaper zone, skin), so that this is rather another dysbiosis site ? Table 1, do you have co-collected samples at other sites in the same cases, or in the human milk ?

Perhaps repetitive to previous paper, but some information on NG tube handling (? Gloves, etc) and EOS/LOS antibiotic modalities is of add on value.
